# Dopamine Pharmacodynamics: New Insights

**DOI:** 10.3390/ijms25105293

**Published:** 2024-05-13

**Authors:** Fulvio Lauretani, Francesco Giallauria, Crescenzo Testa, Claudia Zinni, Beatrice Lorenzi, Irene Zucchini, Marco Salvi, Raffaele Napoli, Marcello Giuseppe Maggio

**Affiliations:** 1Geriatric Clinic Unit, Geriatric-Rehabilitation Department, University Hospital, 43126 Parma, Italy; crescenzo.testa@unipr.it (C.T.); claudia.zinni@unipr.it (C.Z.); beatrice.lorenzi@unipr.it (B.L.); irene.zucchini@unipr.it (I.Z.); marco.salvi@unipr.it (M.S.); marcellogiuseppe.maggio@unipr.it (M.G.M.); 2Cognitive and Motor Center, Medicine and Geriatric-Rehabilitation Department of Parma, University-Hospital of Parma, 43126 Parma, Italy; 3Department of Translational Medical Sciences, “Federico II” University of Naples, via S. Pansini 5, 80131 Naples, Italy; francesco.giallauria@unina.it (F.G.); raffaele.napoli@unina.it (R.N.)

**Keywords:** reward system, dopamine, neuromodulation, addiction, neurotoxicity

## Abstract

Dopamine is a key neurotransmitter involved in physiological processes such as motor control, motivation, reward, cognitive function, and maternal and reproductive behaviors. Therefore, dysfunctions of the dopaminergic system are related to a plethora of human diseases. Dopamine, via different circuitries implicated in compulsive behavior, reward, and habit formation, also represents a key player in substance use disorder and the formation and perpetuation of mechanisms leading to addiction. Here, we propose dopamine as a model not only of neurotransmission but also of neuromodulation capable of modifying neuronal architecture. Abuse of substances like methamphetamine, cocaine, and alcohol and their consumption over time can induce changes in neuronal activities. These modifications lead to synaptic plasticity and finally to morphological and functional changes, starting from maladaptive neuro-modulation and ending in neurodegeneration.

## 1. Introduction

Since the study by Carlsson and colleagues [1,2], dopamine, and the neurotransmitter system in which it is involved, has become the subject of numerous studies. They discovered it in the late 1950s, and its discovery was worth the Nobel Prize for Physiology and Medicine in 2000.

Dopamine and the dopaminergic system were initially only associated with motor function “as an antagonist of the akinetic effects induced by reserpine”. These systems over the years have become the main field of studies to frame Parkinson’s disease, schizophrenia, drug addiction, attention deficit hyperactivity disorder, mood disorders, and cognitive disorders both in initial and advanced stages [3].

In addition to clinical studies, there are numerous preclinical insights derived from the study of dopamine receptors that modulate fundamental aspects of human physiology [4].

Our working group previously investigated the function of the dopaminergic system, also in its molecular components [5]. The novelty and main purpose of this review is to integrate preclinical and clinical knowledge regarding the dopaminergic system and to explore dopaminergic neuromodulation capable of modifying neuronal architecture.

The most recent acquisitions in this field indicate the need to move from monophasic receptor–neurotransmitter vision. The dopaminergic system is involved in many domains of both motor and cognitive encephalic functions. In the microdomains that underlie the dopaminergic function as a whole, there are much more complex connections that contribute to the formation of a “dopaminergic layer” within the brain. This work intends to review the knowledge of the topic and lay the foundations for future research regarding the prospects of new acquisitions in this field. As a consequence, we speculate the basis for identifying drugs that are modifiers of the pathological trajectories of neurodegenerative diseases.

## 2. The Dopamine System

Although dopaminergic neurons represent less than 1% of cell subtypes in the brain [6], dopamine embodies 80% of catecholamines in the central nervous system (CNS) [7]. Dopaminergic neurotransmission is involved in pleasurable reward, behavior, cognition, attention, learning, sleep, and emotion [8]. It is also involved in motor function since dopaminergic neurons correspond to 3–5% of substantia nigra [6]. Its complex role provides an explanation to its implication in many different diseases such as Parkinson’s, schizophrenia, attention deficit hyperactivity disorder (ADHD), autism spectrum disorders, obsessive compulsive disorder, substance dependency, and others [9]. Such heterogeneity of physiological functions and involvement in the pathophysiology of human disorders is entailed by an equal heterogeneity of dopamine receptors and brain areas in which these receptors are expressed. In turn, these brain areas are involved in a dopaminergic macrosystem that comprehends areas that are spatially very distant from each other [3].

There are five subtypes of dopamine receptors described, all belonging to the G protein–coupled receptors (GPCR) family. These five receptors (D1 dopamine receptor, D2 dopamine receptor, D3 dopamine receptor, D4 dopamine receptor, and D5 dopamine receptor) are further divided into two main subclasses. The “D1-like” group includes the D1 dopamine receptor and D5 dopamine receptor, while the “D2-like” group includes the D2 dopamine receptor, D3 dopamine receptor, and D4 dopamine receptor. Structure and sensitivity to drugs are known to be similar in the same subclasses [10].

The signal transduction downstream of dopamine receptors makes use of Gs and Gi proteins, typical of the GPCR family. The major second messenger in the dopamine receptor signaling cascade is cyclic adenosine monophosphate (cAMP) [11].

The final effect of dopaminergic stimulation on the target neuron depends on the receptor subtype and on the capacity to increase or decrease the intracellular cAMP concentration of that given neuron [12]. At a general level, the effect of D1-like receptors is both excitatory (if coupled to the opening of sodium channels) and inhibitory (if coupled to the opening of potassium channels). The associated signal transduction is generally implicated in postsynaptic inhibition. On the other hand, the effect of D2-like receptors is inhibitory with regard to the potential of the membrane, both at the presynaptic and postsynaptic levels [13]. D1-like and D2-like receptors differ in several aspects: the response to the different agonists and antagonists and the effectors downstream of the cascade that underlie this response, the distribution in the different brain areas, and the mechanisms in which the release of dopamine and the signal cascade are interrupted [14]. D1 dopamine receptors are the most expressed dopamine receptor in the central nervous system [15].

Dopamine receptors are mainly expressed in the mesolimbic, nigrostriatal, and mesocortical areas, particularly in the substantia nigra, olfactory nucleus, and nucleus accumbens. There is evidence of their expression in the basal nuclei system (caudate, putamen, and striatum) [15]. Their stimulation regulates the voluntary control of movements, the sense of satiety and hunger, attention, affective behavior, sexual behavior, learning, and working memory [16]. D2 dopamine receptors are the second most abundant dopaminergic receptor expressed in the central nervous system. They are also expressed in the substantia nigra, olfactory bulb, ventral tegmental area (VTA), and nucleus accumbens. They are involved in working memory, reward-motivation functions, and a lot of parasympathetic functions [17]. D3, D4, and D5 dopamine receptors are expressed at significantly lower levels. The D3 dopamine receptor is expressed only in the CNS and not outside it. It is found in the olfactory bulb and nucleus accumbens and is involved in the modulation of neuroendocrine function, emotions, and drug addiction [18]. The D4 dopamine receptor is less expressed in the CNS, mostly found outside. In CNS, it is expressed in the substantia nigra, hippocampus, amygdala, thalamus, hypothalamus, and frontal cortex. It is involved in modulations of cognitive functions but primarily in regulations of sympathetic and parasympathetic functions [19].

Finally, the D5 dopamine receptor is also expressed in the substantia nigra, hypothalamus, dental gyrus, and hippocampus and is involved in the pain process, regulation of endocrine function, affective behavior, and hypertension [20]. Besides structural and expression differences, a functional difference should also be noted. Preclinical receptor affinity studies have highlighted profound differences in terms of affinity for dopamine between dopamine receptors: receptors belonging to the D2-like receptor family have an affinity for dopamine that is 10 to 100 times greater than the receptors of the D1-like group. This difference is due to the ability of the receptors to recognize a different mode of dopamine release which underlies different neurobiological meanings: the D1-like receptors are activated by high concentrations of released dopamine, while the D2-like receptors are activated by low levels of dopamine [6,21,22]. It is evident that this variety of integration of the dopaminergic signal underlies a greater characterization of the information conveyed via the different modes of firing pattern at the level of the dopaminergic circuits [4]. It is important to underline that dopaminergic neurotransmission is not only characterized by biological differences in the receptors and by the electrophysiological properties of dopaminergic neurons but also by the postsynaptic effects of dopamine and its clearance at the extracellular level [23]. Evolution has selected ways to carefully control the extracellular handling of dopamine. Dopamine uptake is controlled by a specific transporter (DAT), the target of drugs of abuse such as amphetamine and cocaine. There are other regulators of the extracellular concentration of dopamine such as monoamine oxidases, which degrade dopamine, or catecholoxy methyltransferases, which deactivate it. There are also monoamine transporters selectively capable of transporting dopamine as well [24]. It is known that alterations in this careful dopamine-handling system can be the predisposing cause of psychiatric disorders [14].

Dopamine neurons represent approximately 1% of the neurons present within the human brain, and the majority of these are located at the level of the ventral midbrain, in particular at the level of the substantia nigra [25]. Overall, in the human brain, 135 thousand dopaminergic neurons have been described at the level of the substantia nigra and 35 thousand at the level of the ventral tegmental area [7]. The projections arising from these brain areas constitute the fundamental framework of the human dopaminergic system. Two main mechanisms modulate dopamine release and its extracellular levels: phasic and tonic. Phasic dopamine neuronal firing leads to a fast and transient increase in dopamine concentration while tonic release generates a milder and less intense increase [26,27]. Four main functionally and anatomically interconnected pathways of the dopaminergic system are as follows: the mesolimbic, the nigrostriatal, the mesocortical, and the tuberoinfundibular pathways. The first two are markedly dopaminergic, and the last two also make use of the contribution of serotonergic transmission [3]. The mesolimbic pathway arises from the ventral tegmental area (VTA) and projects to the amygdala, pyriform cortex, lateral septal nuclei, and the nucleus accumbens. It is deeply connected to pleasure in the brain: pleasurable situations and experiences such as food, sex, or drugs of abuse stimulate mesolimbic dopamine release. Dysfunctions in this pathway lead to craving behavior in drug addiction and cognitive impairment [28,29,30,31]. The nigro-striatal pathway originates from substantia nigra (pars compacta) and spreads in the basal ganglia (caudate nucleus and the putamen). Nigro-striatal dopamine is central in motor control, and it is also implicated in central pain modulation. Blockage of nigro-striatal dopamine receptors is the cause of the extrapyramidalic effects of antipsychotic drugs [32,33,34,35]. The mesocortical pathway, which originates from VTA and projects to the frontal cortex and septohippocampal regions, modulates cognitive and emotional behavior and processes. It also describes its associated with the glutamatergic pathway [36,37,38]. The tubero-infundibular pathway, in which fibers originate from the hypothalamus (arcuate and paraventricular nuclei) and project to the median eminence of the pituitary gland, is related to the lactation phenomenon via prolactin inhibition [39,40]. In conclusion, various parts of the dopamine system are linked both anatomically and physiologically, and this connection allows for appropriate functionality. On the other hand, any alteration of these networks could be a predisposition to various diseases [41,42,43].

## 3. Dopamine System and Human Disease

The dopamine system is implicated in the development and progression of various human disorders, altogether in the up- and downregulation of its pathways (as shown in Figure 1).

Schizophrenia is a relatively common and debilitating psychiatric disease with a heterogenous combination of three types of symptoms (positive, negative, and cognitive). Positive symptoms include hallucinations, disorganized speech, and behavior; negative symptoms include impaired motivation and social withdrawal, and cognitive symptoms include memory dysfunction. Three different hypotheses, namely neurodevelopment, glutamate, and finally dopamine have been proposed to explain the physiopathology of this disorder. The dopamine hypothesis, developed in the 1960s, evolved from two observations: dopamine blockers could reduce psychotic symptoms, and amphetamines increase dopamine, exacerbate schizophrenic symptoms. Patients with schizophrenia show higher levels of post-mortem levels of dopamine and metabolites corroborating the dopamine hypothesis [44,45]. Dopamine plays a central role in the pathogenesis of ADHD [46]. Parkinson’s disease is the most common type of Parkinsonism. Parkinsonism refers to a group of neurological disorders with Parkinson’s disease-like movements and problems (rigidity, bradykinesia, and tremor). Other types of Parkinsonism include neurodegenerative disease, drug-induced Parkinsonism, and vascular parkinsonism. A key feature in the pathophysiology of Parkinsonism is a loss of dopaminergic functions [47,48,49]. Dopamine is also implicated in mood disorders. In particular, the decrease in dopamine activity has been involved in depression while the increase in dopamine levels exacerbates mania and manic episodes [50]. Dopaminergic system dysfunction is linked to the physiopathology of purely neurodegenerative diseases. It is well known that this dysregulation of the dopaminergic system is found both in early (motoric–cognitive risk syndrome) [5] and advanced stages of dementia (such as Alzheimer’s disease) [51,52].

Dysfunctions of dopamine signaling are central in Huntington’s disease, an autosomal dominant disorder caused by the repletion of CAG of the huntingtin gene. This disease, which is a typical movement disorder, may later present cognitive and psychiatric disturbances. Neurodegeneration occurs in the caudate and putamen [53,54,55,56,57].

Dopamine is also a key player in substance use disorder. Addiction is characterized by compulsive drug intake, the disability to restrict drug intake, and the withdrawal syndrome. Alterations of synaptic plasticity in the mesolimbic pathway of the dopamine system lead to the dysfunction of the reward system, a pathognomonic feature of drug addiction. Those who are drug-addicted show alterations in the expression of D2 dopamine receptors (low D2 dopamine receptor levels) in the striatal area. There is clinical and preclinical evidence for low levels of D2 dopamine receptors in patients with obesity where there is radiologic evidence (PET) of less availability of D2 dopamine receptors in the same areas. Moreover, preclinical trials have investigated the modulation of neuroplasticity in the dopaminergic system. Drug addiction is promoted and carried by the transient downregulation of autoreceptor sensitivity and D1 dopamine receptor superactivity [58,59,60,61,62,63,64,65,66,67].

## 4. Dopamine as a Neuromodulator

Drug consumption and its transition to addiction are driven not only by drugs‘ pharmacological effects but also influenced by genetic variability, social environments and social support, childhood exposure, and drug accessibility [25]. Drugs of abuse, via their pharmacological effects, increase the release of dopamine in the Nucleus Accumbens mimicking the phasic dopamine neuronal firing. This fast and transient peak in concentration of dopamine levels stimulates D1 dopamine receptors and activates the direct striatal pathway. The subject of using drugs as a reward perceives this phenomenon. When the drug-induced release of dopamine is sufficiently large and fast, it could stimulate both D1 and D2 dopamine receptors, leading to activation of the direct pathway and inhibition of the indirect pathway. While the direct pathway leads to reward, the indirect pathway is associated with punishment. We have three circuits involved in the addiction: the mesolimbic (NAc, amygdala, and hippocampus), the mesocortical (cingulate gyrus and orbito-frontal cortex), and the nigro-striatal circuit (dorsal striatum). The mesolimbic circuit is associated with reward via the stimulation of D1-receptors in NAc. The mesolimbic circuit is also associated with association learning and conditioning explaining drug-related memories and conditioned responses. The reinforcing effect of drugs of abuse is linked to these conditioned responses that shape the expectation that the subject has of the drug effect. It is also well-known that drug consumption is perceived as more pleasurable when the subject expects to receive it. This mechanism is also demonstrated in preclinical trials [68].

The mesocortical circuit is associated with compulsive behavior resulting in poor inhibitory control. Activation of this circuit is reported to be involved during craving. The nigrostriatal circuit, particularly the dorsal striatum, is involved in habit formation. The involvement of different regions, from reward mechanisms in the mesolimbic circuit to habit formation in the nigrostriatal circuit, explains the transition from controlled consumption to addiction [69]. Addiction processes are complex, and we are aware of the importance of a plethora of other neurotransmitters involved (such as glutamate, GABA, norepinephrine, and serotonin), but we would like to explore the relevance of dopamine and dopamine system in drug addiction as a model not only of neurotransmission but also of neuromodulation. It is known that neurotransmission is influenced by factors acting on the dopamine system. Drugs of abuse and their consumption over time can induce change in neuronal activities leading to synaptic plasticity and finally to morphological and functional changes. As demonstrated by Robinson and colleagues in a preclinical in vivo trial, acute administration of cocaine in naive animals induces transcription and epigenetic modulatory events involving genes like fosB, FosB, NFB, CdK5, and MEF2, all associated with regulation and signaling.

This process is reported to happen 1 h after the administration of the drug [70]. If cocaine administration stays consistent, these changes could pass from transient to permanent. This is further shown in a previous preclinical trial where the medium term of assumption (days) reveals changes in neuronal signaling (i.e., long-term potentiation). In the long term (months), there are neuroadaptation phenomena like increased dendritic spine density that lead to cytoskeletal and circuit remodeling.

Finally, in chronic drug exposure (months to years), this remodeling and rearrangements are linked to the persistence of compulsive behaviors in addiction [71]. Because of the long-term consumption of cocaine in terms of remodeling and rearrangements, not only functional but also morphological, we could refer to a clinical trial of 29 subjects (16 naive and 13 cocaine-addicted). The study demonstrated dissimilarities in brains of patients with a cocaine addiction and naive subjects; differences were detected in areas that control decision-making and behavior inhibition, typical of addiction behavior as shown in imaging studies (MRI) that documented decreased gray matter in the orbital–frontal cortex (OFC) [72]. Many factors sustain neuromodulation, as shown in Figure 2 [73,74].

Amphetamine and methamphetamine have an affinity for DAT and can evoke responses like that of cocaine, even if the pharmacokinetics are different [75].

Even alcohol consumption differently influences levels of dopamine. In fact, while drinking acutely increases dopamine levels [76], the brain adapts to the dopamine overload with continued alcohol use, especially at high dosages (for example more than six drinks per day). Under chronic alcohol exposure, there is less production of neurotransmitters, reduction in the number of dopamine receptors in the body, and increasing dopamine transporters, which carry away the excess dopamine [77]. There is a lot of evidence indicating that alcohol can increase brain levels of monoamines independently of the action on transporters (like DAT) [78]. Furthermore, it has been demonstrated that alcohol consumption is able to modify both the gene expression of neurons that make up the nuclei of the reward system [79] and to modify their synaptic plasticity [80]. It is therefore possible to hypothesize that even in response to chronic alcohol consumption, maladaptive neuromodulatory responses can be triggered which can then generate behavioral disorders like addiction. This hypothesis is confirmed by preclinical evidence indicating that the reward system is strongly influenced by alcohol consumption. In fact, alcohol consumption triggers anticipatory responses, which are specific to the reward system, and leads to a modification of behavior with a shift towards aggression [81]. Furthermore, it has been shown that alcohol consumption can induce maladaptive decision-making processes supported by a dysfunction of the dopaminergic system and in particular of the ventral tegmental area [82]. These maladaptive neuro-modulatory processes occur more easily the earlier the contact with alcohol [83].

It is also important to underline that the neuromodulation of the dopaminergic system triggered by alcohol consumption is also due to its action on the endogenous opioid system [84]. Numerous evidence highlights how chronic alcohol consumption leads to a hypodopaminergic state in the nucleus accumbens and can cause striatal dysfunction [85,86]. This evidence is so important that clinical trials have begun regarding the possible use of partial agonists of endogenous opioid receptors in alcohol use disorders [87].

The opioid system is also capable of having a neuromodulatory effect on the reward system [88]. Substances such as amphetamine and cocaine also elicit their effect via interaction with the endogenous opioid system, triggering maladaptive neuromodulation processes in the reward system [89,90]. Morphological changes in brain structure are established after a period of years, therefore after an addiction has been established. As mentioned above, the transition from occasional consumption to addiction is strictly influenced by genetic factors. Genetic polymorphism can play a significant role in interindividual differences in addition to risk, and some studies show that nearly 50% of addiction risk is determined by one’s genetic profile [91,92]. The predisposition to taking a substance of abuse is also influenced by genotype. In a study on 1.2 million subjects, it was highlighted how different stages of tobacco addiction and the time spent in a stage (initiation, cessation, and heaviness) were influenced by genetic loci that were actively involved in dopaminergic transmission [93].

Other apparently less influential factors can modify the reward system. For example, the pleiotropic effects of physical exercise are not only expressed at a systemic level [94] but also with a direct action on the reward system. It has been abundantly demonstrated that physical exercise is able to act with a neuromodulating action on the reward system, so much so that physical exercise protocols have been proposed in the treatment of addictions and behavioral disorders linked to substance abuse [95,96,97,98].

All the studies presented in this review show the enormous progress made in understanding the functions of dopamine. Dopamine is changing its role from a simple neurotransmitter to a neuromodulator capable of modifying the function and response to stimuli of numerous brain areas [99,100]. Numerous aspects of dopaminergic function have been studied in relation to not simply receptor activities. There is evidence of the typical firing pattern in relation to prediction error [101] and the anatomical and functional connection with other neuroreceptor systems such as the serotoninergic and endocannabinoid systems [102]. Both anatomical (coexistence of different neuronal types in certain brain areas) and functional interactions between different neurotransmitter systems that integrate dopaminergic function explain the involvement of dopamine in high cognitive functions [103]. The novelty, recently supported by studies, is that the high cognitive functions are not only supported by interconnected functional macrodomains [5] but also by functional microdomains strongly and structurally integrated [103,104]. Even if the molecular bases of these interactions are to be fully elucidated, it seems that neuromodulation mediated by “classic” neurotransmitters can modify the biophysical properties of action potentials. By these mechanisms, the integration of the neuronal signals of cerebral areas apparently functionally segregated is reached [105,106,107]. An important role in these neuromodulation processes is played by dopamine [108,109]. Very recently, it has been shown that the topographic distribution of dopamine receptors constitutes a functional architecture within the brain and is one of the main determinants of high cognitive functions. The discovery of this “dopaminergic layer” in the topographical and functional organization of the brain lays the foundations for innovative approaches to numerous psychiatric and neurodegenerative diseases [104]. This dopaminergic layer is also supported by a functional organization and a specific receptor density of dopamine receptors within the different brain areas as confirmed by recent studies [110,111,112,113].

We have already seen how minimal genetic differences in receptor structure or expression can be a predisposition to the establishment of pathological addictions [93]. It has been shown that individual differences in dopaminergic receptors can influence brain architecture and connections within the dopaminergic layer [114,115]. The receptor organization is not only closely connected to major cognitive functions but also in different periods of life, there are shifts in their density. These changes can lay the foundation for innovative approaches to the difficult learning of cognitive decline in the elderly [116,117]. This recent evidence integrates knowledge of dopaminergic function and the ability of dopamine to function as a modulator of synaptic plasticity in both cognitive and motor domains of the brain [9,118,119].

A very recent trial published on April 3 demonstrates that the action of Lixisenatide, a glucagon-like peptide-1 receptor agonist used for the treatment of diabetes, is somehow neuroprotective in patients with early onset of Parkinson’s disease [120]. These recent data suggest the usefulness of GLP-1 agonists as possible disease modifiers in neurodegenerative disease [121,122,123]. These drugs not only show neuroprotective function but can also contribute to the symptomatic relief of patients [124]. Although the molecular pathways underlying these neuroprotective effects are yet to be completely elucidated, numerous preclinical studies indicate that glucagon-like peptide-1 receptors may directly regulate components of the dopaminergic system [125].

These innovative and recent acquisitions, combined with the study of the topography of the brain dopaminergic system, are potentially very important in the reconceptualization and treatment of addictions, psychoses, and neurodegenerative diseases.

## 5. From Bad Neuromodulation to Neurotoxicity

Dysfunctional dopaminergic neurotransmission has been associated with cell death. An intrastriatal injection of dopamine can cause DNA damage and apoptosis in the rat brain [126,127]. Furthermore, in preclinical models, striatal cells undergo cell death when exposed to high concentrations of dopamine [128,129,130,131]. Dopamine was found to be toxic, at different concentrations, and also in other cytotypes [132,133,134,135,136,137]. Dopamine metabolism generates numerous reactive oxygen species, and this has been indicated as the main pathogenetic step in dopamine-induced neurodegeneration [138,139,140].

Toxic effects have also been found at the mitochondrial level [141]. But the toxic effects of dopamine are not only indirect but also direct: it seems that the interaction with the D1 dopamine receptor is responsible for the neurodegeneration caused by dopamine [142]. A complementary experiment demonstrated that the toxic effects of dopamine were completely reversed in the presence of antioxidant substances and hyper-specific D1 dopamine receptor antagonists [143]. Analyzing the effects of methamphetamine highlights a significant increase in dopamine levels in the brain [144]. Furthermore, there is preclinical evidence of the neurotoxic effects of methamphetamine [145,146] due to a dysfunction of apoptosis, stress of the endoplasmic reticulum, and also to interaction with the dopamine D1 dopamine receptor. Treatment with a hyper-specific D1 dopamine receptor antagonist blocks the neurotoxic effects of methamphetamine [147,148]. Neurodegenerative effects are not only ascribed to methamphetamine but also to other substances of abuse and alcohol [149,150,151] (Figure 3).

Chronic alcohol use, as with methamphetamine, alters neuron gene expression and activates gene patterns that promote mitochondrial dysfunction [152]. Furthermore, it has been widely described that chronic alcohol use promotes neuroinflammation and consequently neurodegeneration [153,154,155,156]. In detail, the acute assumption of alcohol, like most addictive drugs, activates the mesolimbic dopamine system and releases dopamine [157], while chronic exposure to ethanol reduces the baseline function of the mesolimbic dopamine system. However, the molecular mechanisms underlying ethanol’s interaction with this system remain to be elucidated, given the complexity of the actions of alcohol on nervous systems in general and on the dopamine system in particular [158]. Alcohol is even associated with macroscopic alterations of brain volume, even gray or white matter volume, specifically when alcohol intake is noted as more than two or more units (or drinks) daily. This seems to be protective assuming only one drink is taken daily [159,160].

## 6. Conclusions

The aim of this review is to analyze dopaminergic transmission and the effects of the dopaminergic pathway as a neuromodulator. There is numerous and increasing evidence indicating that pathological neuromodulation, in subjects genetically predisposed or hyper-exposed to environmental factors, can evolve towards addiction or even neurodegeneration. Thus, we underline the importance of prevention campaigns and further studies to better elucidate the molecular mechanisms involved in pathological neuromodulation in order to identify possible therapeutic targets.

## Figures and Tables

**Figure 1 ijms-25-05293-f001:**
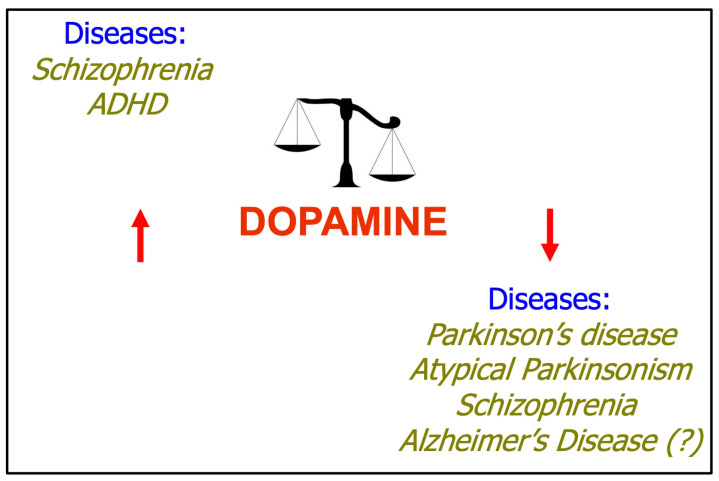
The dopaminergic system can be both activated and hypoactivated in different pathological conditions. In some diseases such as schizophrenia, dopaminergic action can be upregulated or downregulated depending on the brain area considered.

**Figure 2 ijms-25-05293-f002:**
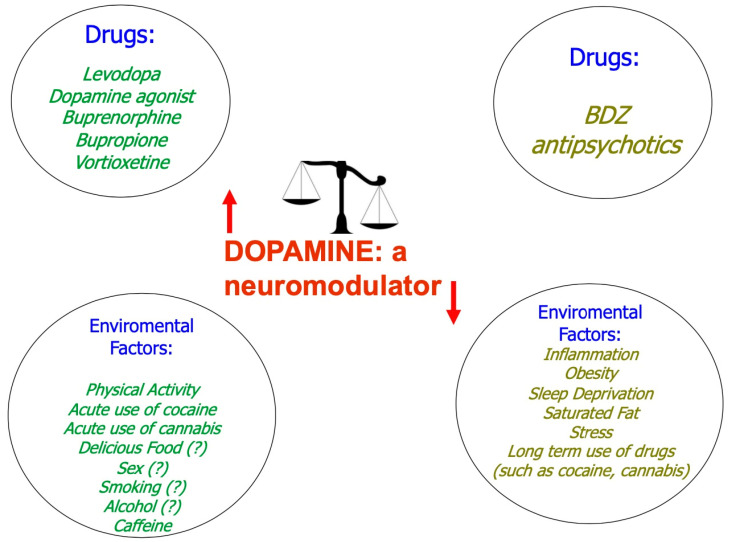
The effect of dopamine is not only receptorial but also extra-receptorial and is in turn regulated by numerous factors.

**Figure 3 ijms-25-05293-f003:**
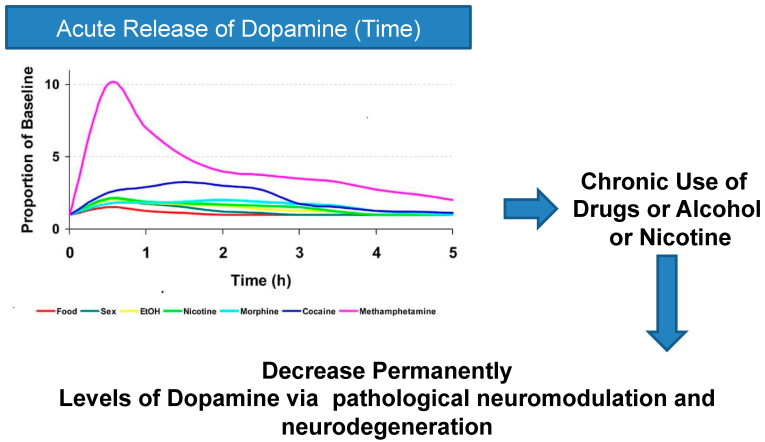
Substances of abuse activate the dopaminergic system in a different way and with a different intensity. This difference is inherent in their different mechanisms of action of substances of abuse. For example, palatable substances also activate the reward circuit through gustatory stimulation.

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
