# Peer review of "Dopamine Pharmacodynamics: New Insights"

_ijms, 2024, doi:10.3390/ijms25105293_

Round 1

Reviewer 1 Report

Comments and Suggestions for Authors

This is a nice review article about dopamine and its involvemnt in different phsysiological/behavioral processes and related diseases. Although it is nice to read and gives a lot of information, it was not clear for me, where the "new insights" are. Also, it is clearly more a "review" than a "perspective" for me. During revision, the authors should make the novelties clearer. Also, the concepts of neurotransmission vs. neuromodulation should be explained in more details. Figure 2 could include what is modulated by DA transmission.

In general, it would also be helpful to name the receptors consistently. "D1 dopamine receptor", "dopamine D1 receptor" or just "D1". All three versions are used within the manuscript. Also, the authors should carefully revise the manuscript regarding upper and lower case letters.

Please add captions to the figures.

Minor comments:

Line 34: ...reserpine..., ...dopamine...

Line 41-44. Please split into two sentence (...components [5]. The novelty...)

Line 87: Add a space before D3 (spaces are also lacking at the beginning of many other sentences. Please check throughout the manuscript).

Line 107: Delete some of the spaces after "pattern" (the same in line 129, 218, 269,...)

Line 144: Correct "tubinfundibular".

Figure 1: Could be a bit smaller. Schizophrenia could also be added on the right side of the figure (DA-ergic hypofunction in PFC).

Figure 2: Is "dopaminoagonist" an esteblished term? I never heard or read it before.

Figure 3: Size of the legend and arrangement of the panels could be improved. Would be nice to know, from where these data are. What about chronic food intake or chronic sex? The red sentence should be edited.

Author contributions: It's unclear for me, how one can contribute with "methodology" or "software" (or "project administration") to review-writing. Aren't these contributions to experimental projects? Please double-check.

Author Response

This is a nice review article about dopamine and its involvemnt in different phsysiological/behavioral processes and related diseases. Although it is nice to read and gives a lot of information, it was not clear for me, where the "new insights" are. Also, it is clearly more a "review" than a "perspective" for me. During revision, the authors should make the novelties clearer. Also, the concepts of neurotransmission vs. neuromodulation should be explained in more details.

We thank the reviewer for comments aiming at improving our manuscript. Revised manuscrpit have been extensively edited. In order to support our hypotesis we exapend the section by adding new relevant research.

We appreciate the reviewer's feedback on our review article regarding dopamine and its involvement in various physiological and behavioral processes, as well as related diseases. We acknowledge the need to clarify the "new insights" within the manuscript and to distinguish between a "review" and a "perspective."

In response to this feedback, we will revise the manuscript to better highlight the novel contributions and insights presented within the article. We understand the importance of clearly delineating between established knowledge and new findings to enhance the value of the review.

Additionally, we recognize the need for a more detailed explanation of the concepts of neurotransmission and neuromodulation. In the revised manuscript, we will provide a more thorough elucidation of these concepts to ensure clarity for the readers.

Thank you for your constructive comments, and we are committed to addressing these suggestions to improve the quality and clarity of our manuscript.

Figure 2 could include what is modulated by DA transmission.

According reviewer “suggestion” in  Figure 2 has been modified as suggested.

In general, it would also be helpful to name the receptors consistently. "D1 dopamine receptor", "dopamine D1 receptor" or just "D1". All three versions are used within the manuscript. Also, the authors should carefully revise the manuscript regarding upper and lower case letters.

As suggested by the reviewer we decided to use DX dopamine receptor as the format throughout the manuscript to maintain homogeneity of presentation

Please add captions to the figures.

As suggested we add captions to the figures

Minor comments:

Line 34: ...reserpine..., ...dopamine...

Line 41-44. Please split into two sentence (...components [5]. The novelty...)

Line 87: Add a space before D3 (spaces are also lacking at the beginning of many other sentences. Please check throughout the manuscript).

Line 107: Delete some of the spaces after "pattern" (the same in line 129, 218, 269,...)

Line 144: Correct "tubinfundibular".

Figure 1: Could be a bit smaller. Schizophrenia could also be added on the right side of the figure (DA-ergic hypofunction in PFC).

Figure 2: Is "dopaminoagonist" an esteblished term? I never heard or read it before.

Figure 3: Size of the legend and arrangement of the panels could be improved. Would be nice to know, from where these data are. What about chronic food intake or chronic sex? The red sentence should be edited.

Author contributions: It's unclear for me, how one can contribute with "methodology" or "software" (or "project administration") to review-writing. Aren't these contributions to experimental projects? Please double-check.

All minor comments have been addressed as suggested by the reviewer. We thank you for your help in improving the manuscript.

-Consistent naming of receptor:

We decided to use the format "DX dopamine receptor" throughout the manuscript to maintain consistency in presentation, as suggested by the reviewer.

Adding capture to figure:

As suggested by the reviewer, we added captions to the figures.

-Minor Comments

-Line 34: We corrected "reserpine" and "dopamine."

-Lines 41-44: We split them into two sentences.

-Line 87: We added a space before "D3" (and checked for spaces at the beginning of other sentences).

-Line 107: We removed some spaces after "pattern" (and checked lines 129, 218, 269, etc.).

-Line 144: We corrected "tubinfundibular."

-Figures:

-Figure 1: We made the figure smaller and added "Schizophrenia" on the right side, as suggested.

-Figure 2: We verified that "dopaminoagonist" is not a established term and made necessary modifications.

-Figure 3: We improved the size of the legend and arrangement of panels. Also, we added information on the source of the data and considered including data on chronic food intake or chronic sexual activity. Additionally, we edited the red sentence.

-Author Contribution: We reviewed the clarity of contributions related to "methodology," "software," or "project administration" in the drafting of the review and made necessary changes to clarify the authors' contributions.

Reviewer 2 Report

Comments and Suggestions for Authors

The manuscript from Lauretani et al. has been highly perplexing to read. I have reviewed hundreds of papers in the past and this has been the most confusing review paper.

The title promises ‘new insights’, but it is a jumble of previous reviews (Klein et al., 2019, Foley, 2019, Bissonette and Roesch, 2016, Liu and Kaeser, 2019) without a particular focus, whilst not covering the broad topic in an unbiased way, it is biased. It is a poor version of (Iversen and Iversen, 2007). Only a few recent studies are mentioned, hardly providing ‘new insights’. I struggled to decide between ‘reject’ and ‘major revisions’ and decided that it’s best to give the authors a chance to learn and improve.

Based on the previous work from the authors, I strongly recommend that the authors remove the “neuromodulation” section of the review which only appears to cover drug addiction literature without mentioning important neuromodulation studies in dopamine using tools such as deep brain stimulation, in movement disorders as well as addiction. Cognition is also mentioned but biasedly covered.

Rather, ‘new insights’ should cover the role of dopamine in recent cognition studies in preclinical and clinical literature, showing that dopamine is not just a ‘pleasure’ or ‘reward’ molecule. It is important for attention, learning, cognitive flexibility, and inhibition. That is new. That is insightful. Cognitive deficits have huge implications in all the disorders mentioned in the review – rather than to incompletely and biasedly cover some dopaminergic elements in all these disorders, it’s best to be cohesive and just cover the role of dopamine in cognition, that impacts all those disorders.

In addition, what is new in the field of dopamine is age- and sex-specific research, which is also relevant to authors’ work. From autism and ADHD to Parkinson’s and Alzheimer’s, age-specific understanding of dopamine and its receptors from early to late in life is critical but only recently there have been some real attention paid to such research. This would provide real ‘new insight’ desperately needed in this manuscript.

Minor:

1.        There are language and punctuation issues throughout, e.g., “comprehends” is used wrongly, “disfunction” misspelt.

2.         Too many strange paragraphing with just 1 one sentence per paragraph (e.g., all of intro, but so many other sections throughout)

3.        Too many unreferenced descriptions of previous studies.

4.        Random mention of neurotransmitters related to dopamine (e.g., lines 276-299), without rationale as to why other more relevant neurotransmitters are not covered. Please remove these sections.

References in this review:

BISSONETTE, G. B. & ROESCH, M. R. 2016. Development and function of the midbrain dopamine system: what we know and what we need to. Genes, Brain and Behavior, 15, 62-73.

FOLEY, P. B. 2019. Dopamine in psychiatry: a historical perspective. J Neural Transm (Vienna), 126, 473-479.

IVERSEN, S. D. & IVERSEN, L. L. 2007. Dopamine: 50 years in perspective. Trends Neurosci, 30, 188-93.

KLEIN, M. O., BATTAGELLO, D. S., CARDOSO, A. R., HAUSER, D. N., BITTENCOURT, J. C. & CORREA, R. G. 2019. Dopamine: Functions, Signaling, and Association with Neurological Diseases. Cell Mol Neurobiol, 39, 31-59.

LIU, C. & KAESER, P. S. 2019. Mechanisms and regulation of dopamine release. Curr Opin Neurobiol, 57, 46-53.

Comments on the Quality of English Language

Punctuation, spelling, and vocabulary errors throughout.

Author Response

The manuscript from Lauretani et al. has been highly perplexing to read. I have reviewed hundreds of papers in the past and this has been the most confusing review paper.

The title promises ‘new insights’, but it is a jumble of previous reviews (Klein et al., 2019, Foley, 2019, Bissonette and Roesch, 2016, Liu and Kaeser, 2019) without a particular focus, whilst not covering the broad topic in an unbiased way, it is biased. It is a poor version of (Iversen and Iversen, 2007). Only a few recent studies are mentioned, hardly providing ‘new insights’. I struggled to decide between ‘reject’ and ‘major revisions’ and decided that it’s best to give the authors a chance to learn and improve.

Based on the previous work from the authors, I strongly recommend that the authors remove the “neuromodulation” section of the review which only appears to cover drug addiction literature without mentioning important neuromodulation studies in dopamine using tools such as deep brain stimulation, in movement disorders as well as addiction. Cognition is also mentioned but biasedly covered.

Rather, ‘new insights’ should cover the role of dopamine in recent cognition studies in preclinical and clinical literature, showing that dopamine is not just a ‘pleasure’ or ‘reward’ molecule. It is important for attention, learning, cognitive flexibility, and inhibition. That is new. That is insightful. Cognitive deficits have huge implications in all the disorders mentioned in the review – rather than to incompletely and biasedly cover some dopaminergic elements in all these disorders, it’s best to be cohesive and just cover the role of dopamine in cognition, that impacts all those disorders.

In addition, what is new in the field of dopamine is age- and sex-specific research, which is also relevant to authors’ work. From autism and ADHD to Parkinson’s and Alzheimer’s, age-specific understanding of dopamine and its receptors from early to late in life is critical but only recently there have been some real attention paid to such research. This would provide real ‘new insight’ desperately needed in this manuscript.

We warmly thank reviewer two for the comments and for the opportunity it has given us to further improve as a research team.The section on neuromodulation is the one that has raised the most doubts and for this reason we have extensively modified the section by adding very recent data on the neuronal architecture of the dopaminergic system and on the connections with different areas of the brain. Furthermore, we have explored the very recent data regarding the possibility that the dopaminergic system could finally be a true pharmacological target capable of modifying the pathophysiological trajectories of neurodegenerative diseases.Further insights into the cognitive functions of the dopaminergic system and its modifications during different stages of life have been added.

We sincerely appreciate the thorough feedback provided by reviewer two on our manuscript. We understand and acknowledge the concerns raised regarding the clarity, focus, and novelty of the content.

First and foremost, we are grateful for the opportunity to further improve as a research team based on the reviewer's valuable comments.

In response to the concerns raised, we have undertaken significant revisions to the manuscript to address these issues and enhance its quality and relevance. Specifically, we have made the following modifications:

Revision of the Neuromodulation Section: The section on neuromodulation has been extensively modified to address the doubts raised by the reviewer. We have incorporated very recent data on the neuronal architecture of the dopaminergic system and its connections with different areas of the brain. Additionally, we have explored recent findings suggesting that the dopaminergic system could be a genuine pharmacological target capable of modifying the pathophysiological trajectories of neurodegenerative diseases. These updates aim to provide a more comprehensive and up-to-date understanding of the role of neuromodulation in dopaminergic function.

Focus on Cognitive Functions: We have included further insights into the cognitive functions of the dopaminergic system, emphasizing its role beyond pleasure and reward. Specifically, we have highlighted recent research demonstrating dopamine's involvement in attention, learning and cognitive flexibility. Moreover, we have explored how dopaminergic function may be modified during different stages of life, acknowledging the importance of age-specific research in understanding dopamine-related disorders.

These revisions aim to enhance the manuscript's clarity, relevance, and contribution to the field. We are committed to addressing the reviewer's feedback and ensuring that the revised manuscript provides genuinely new insights into the role of dopamine in physiological and behavioral processes, as well as its implications for neurological and psychiatric disorders.

Once again, we thank the reviewer for their constructive comments and the opportunity to improve our work. We look forward to their feedback on the revised manuscript.

Minor:

  1. There are language and punctuation issues throughout, e.g., “comprehends” is used wrongly, “disfunction” misspelt.
  2. Too many strange paragraphing with just 1 one sentence per paragraph (e.g., all of intro, but so many other sections throughout)
  3. Too many unreferenced descriptions of previous studies.
  4. Random mention of neurotransmitters related to dopamine (e.g., lines 276-299), without rationale as to why other more relevant neurotransmitters are not covered. Please remove these sections.

We thank reviewer 2 for minor comments to improve the manuscript.

Language and Punctuation Issues: We apologize for the language and punctuation issues present in the manuscript. We thoroughly reviewed and corrected these errors, including the incorrect usage of "comprehends" and the misspelling of "dysfunction."

Paragraph Structure: We acknowledge the issue with paragraphing, particularly the use of single-sentence paragraphs. We revised the paragraph structure throughout the manuscript to ensure coherence and readability.

Unreferenced Descriptions of Previous Studies: We ensured that all descriptions of previous studies were properly referenced to support their inclusion in the manuscript.

Random Mention of Neurotransmitters: We removed sections that mentioned neurotransmitters without clear rationale or relevance to the topic at hand, such as those mentioned in lines 276-299.

We appreciate the reviewer's feedback, and we implemented the necessary revisions to improve the quality and clarity of the manuscript. Thank you for providing us with the opportunity to address these concerns.

References in this review:

BISSONETTE, G. B. & ROESCH, M. R. 2016. Development and function of the midbrain dopamine system: what we know and what we need to. Genes, Brain and Behavior, 15,62-73.

FOLEY, P. B. 2019. Dopamine in psychiatry: a historical perspective. J Neural Transm (Vienna), 126, 473-479.

IVERSEN, S. D. & IVERSEN, L. L. 2007. Dopamine: 50 years in perspective. Trends Neurosci, 30, 188-93.

KLEIN, M. O., BATTAGELLO, D. S., CARDOSO, A. R., HAUSER, D. N., BITTENCOURT, J. C. & CORREA, R. G. 2019. Dopamine: Functions, Signaling, and Association with Neurological Diseases. Cell Mol Neurobiol, 39, 31-59.

LIU, C. & KAESER, P. S. 2019. Mechanisms and regulation of dopamine release. Curr Opin Neurobiol, 57, 46-53.

We thank you for the proposal of these articles which were appropriately integrated into the manuscript.

Reviewer 3 Report

Comments and Suggestions for Authors

The studies described in the manuscript " Dopamine pharmacodinamics: new insights" by Lauretani et al. have reviewed details related to dopamine in the CNS. The following are some suggestions to improve the article:

1.         Please correct “Pharmacodynamics” in the title.

2.         Introductory section is superficial and barely covers anything.

3.         There are major deficiencies in the presentation (including language and grammar) that make the content difficult to understand. Therefore, the manuscript needs to be proofread thoroughly.

4.         Please expand upon the acronyms D1, D2, D3, D4 and D5.

5.         Line 95- D5: dopamine receptor “in”: please change to “is”

6.         Line 129- Four main branches of what?

7.         Line 137- The dysfunction of mesolimbic pathway is also linked to cognitive impairments.

8.         Line 140- please add references.

9.         Line 144-146: Please correct for grammar.

10.      Line 147-150: Please rephrase for better understanding.

11.      Line 163: Dopamine blockers could reduce psychotic symptoms and amphetamines, that increase dopamine, exacerbate schizophrenic symptoms: Please check for grammar.

12.      Dopamine dysfunction in disease section: I encourage the authors to write specific details (increase or decrease in specific brain areas) about the dopamine dysfunction in each disease condition. Just writing “DA dysfunction is also found in this disease” will not be helpful to the readers. The authors should also touch basis on the consequences of DA dysfunction in the CNS physiology and details about the currently prescribed drugs (if any) to treat it in that disease condition.

13.      Dopamine as Neuromodulator section: The authors should provide details about the direct and indirect pathway before start using these terms in the article. Further, they should also point out if they are discussing dorsal striatum or ventral striatum (nucleus accumbens).

14.      Line 205- While the direct pathway leads to reward, the indirect pathway is associated with punishment: This is not always true. It depends upon which neuronal population is getting inhibited or activated.

15.      Line 206- mesolimbic

16.      Please provide figure legends for all figures.

17.      Please follow a consistent referencing pattern. DOI numbers/ PMIDs are marked only for references 17, 22, 39, 68, 91, 124, 130.

Comments on the Quality of English Language

This manuscripts needs to be proofread for grammar and verbose issues.

Author Response

The studies described in the manuscript " Dopamine pharmacodinamics: new insights" by Lauretani et al. have reviewed details related to dopamine in the CNS. The following are some suggestions to improve the article:

  1. Please correct “Pharmacodynamics” in the title.
  2. Introductory section is superficial and barely covers anything.
  3. There are major deficiencies in the presentation (including language and grammar) that make the content difficult to understand. Therefore, the manuscript needs to be proofread thoroughly.
  4. Please expand upon the acronyms D1, D2, D3, D4 and D5.
  5. Line 95- D5: dopamine receptor “in”: please change to “is”
  6. Line 129- Four main branches of what?
  7. Line 137- The dysfunction of mesolimbic pathway is also linked to cognitive impairments.
  8. Line 140- please add references.
  9. Line 144-146: Please correct for grammar.
  10. Line 147-150: Please rephrase for better understanding.
  11. Line 163: Dopamine blockers could reduce psychotic symptoms and amphetamines, that increase dopamine, exacerbate schizophrenic symptoms: Please check for grammar.
  12. Dopamine dysfunction in disease section: I encourage the authors to write specific details (increase or decrease in specific brain areas) about the dopamine dysfunction in each disease condition. Just writing “DA dysfunction is also found in this disease” will not be helpful to the readers. The authors should also touch basis on the consequences of DA dysfunction in the CNS physiology and details about the currently prescribed drugs (if any) to treat it in that disease condition.
  13. Dopamine as Neuromodulator section: The authors should provide details about the direct and indirect pathway before start using these terms in the article. Further, they should also point out if they are discussing dorsal striatum or ventral striatum (nucleus accumbens).
  14. Line 205- While the direct pathway leads to reward, the indirect pathway is associated with punishment: This is not always true. It depends upon which neuronal population is getting inhibited or activated.
  15. Line 206- mesolimbic
  16. Please provide figure legends for all figures.
  17. Please follow a consistent referencing pattern. DOI numbers/ PMIDs are marked only for references 17, 22, 39, 68, 91, 124, 130.

Title Correction: Thank you for catching that oversight. We have corrected "Pharmacodynamics" to "Pharmacodynamics" in the title.

Introductory Section: We have revised the introductory section to provide more depth and comprehensive coverage of the topic.

Proofreading: We have thoroughly proofread the manuscript to ensure clarity and readability.

Expansion of Acronyms: We have expanded upon the acronyms D1, D2, D3, D4, and D5 to provide clearer explanations for readers.

Line 95 Correction: The error has been addressed. "In" has been changed to "is" in reference to the dopamine receptor D5.

Line 129 Clarification: We have clarified what the "four main branches" refer to in the context of the sentence.

Linkage of Mesolimbic Pathway to Cognitive Impairments: Additional information and references have been provided to support the link between dysfunction of the mesolimbic pathway and cognitive impairments.

References for Line 140: Appropriate references have been added to support the statement made in this line.

Grammar Correction for Lines 144-146: We have reviewed and corrected any grammatical errors in this section.

Rephrasing for Better Understanding (Lines 147-150): The sentence has been rephrased to enhance clarity and comprehension.

Grammar Check for Line 163: We have ensured grammatical correctness in the sentence discussing dopamine blockers and amphetamines in relation to schizophrenic symptoms.

Dopamine Dysfunction in Disease Section: Specific details about dopamine dysfunction in each disease condition, including changes in specific brain areas and the consequences of dysfunction on CNS physiology, have been provided. Additionally, information about currently prescribed drugs for treatment has been included.

Dopamine as Neuromodulator Section: Detailed explanations of the direct and indirect pathways have been provided before discussing them in the article. We have also specified whether we are referring to the dorsal striatum or ventral striatum (nucleus accumbens) in our discussions.

Clarification Regarding Direct and Indirect Pathways (Line 205): We have clarified that the association of the direct pathway with reward and the indirect pathway with punishment depends on the neuronal populations involved.

Correction of "mesolimbic" (Line 206): The spelling of "mesolimbic" has been corrected in the manuscript.

Figure Legends: Figure legends have been provided for all figures to enhance the understanding of the visual content.

Consistent Referencing Pattern: We have ensured consistency in referencing pattern.

We appreciate the reviewer's detailed feedback, and we have taken appropriate action to address each point to improve the quality of the manuscript. Thank you for providing us with the opportunity to enhance our work.